# HDX-MS reveals nucleotide-dependent, anti-correlated opening and closure of SecA and SecY channels of the bacterial translocon

Zainab Ahdash[1†], Euan Pyle[1,2†], William John Allen[3], Robin A Corey[4], Ian Collinson[3]*, Argyris Politis[1]*

[1]Department of Chemistry, King's College London, London, United Kingdom; [2]Department of Chemistry, Imperial College London, London, United Kingdom; [3]School of Biochemistry, University of Bristol, Bristol, United Kingdom; [4]Department of Biochemistry, University of Oxford, Oxford, United Kingdom

**Abstract** The bacterial Sec translocon is a multi-protein complex responsible for translocating diverse proteins across the plasma membrane. For post-translational protein translocation, the Sec-channel – SecYEG – associates with the motor protein SecA to mediate the ATP-dependent transport of pre-proteins across the membrane. Previously, a diffusional-based Brownian ratchet mechanism for protein secretion has been proposed; the structural dynamics required to facilitate this mechanism remain unknown. Here, we employ hydrogen-deuterium exchange mass spectrometry (HDX-MS) to reveal striking nucleotide-dependent conformational changes in the Sec protein-channel from *Escherichia coli*. In addition to the ATP-dependent opening of SecY, reported previously, we observe a counteracting, and ATP-dependent, constriction of SecA around the pre-protein. ATP binding causes SecY to open and SecA to close; while, ADP produced by hydrolysis, has the opposite effect. This alternating behaviour could help impose the directionality of the Brownian ratchet for protein transport through the Sec machinery.
DOI: https://doi.org/10.7554/eLife.47402.001

*For correspondence:
ian.collinson@bristol.ac.uk (IC);
argyris.politis@kcl.ac.uk (AP)

†These authors contributed equally to this work

Competing interests: The authors declare that no competing interests exist.

## Introduction

Protein transport across and into the membranes that surround and sub-divide cells is essential for life. The ubiquitous Sec translocon performs this task in the plasma membrane of bacteria and the endoplasmic reticulum of eukaryotes. A protein-channel through the membrane is formed by the conserved hetero-trimeric core-complex: Sec61αβγ in eukaryotes, and SecYEG in archaea and bacteria (*Hartmann et al., 1994*). In bacteria, protein secretion is important for cell envelope biogenesis and maintenance, as well as for the delivery of adherence and pathogenic effector proteins to the cell surface. During protein secretion, SecYEG engages with the cytosolic motor ATPase SecA (*Lill et al., 1989*), and together they pass pre-proteins with a short N-terminal cleavable signal sequence across the membrane, whilst still in an unfolded conformation (*Hartl et al., 1990*; *Arkowitz et al., 1993*).

A structure of the complete SecA-SecYEG complex has been determined with an ATP analogue (ADP-BeF$_x$; PDB code 3DIN), as well as with a short pre-protein mimic (*Zimmer et al., 2008*; *Li et al., 2016*) (PDB code 5EUL; modified in *Figure 1a*). The key regions of SecA and SecYEG are highlighted in the context of this model, shown by orthogonal views for clarity (*Figure 1a,b*). No structure of the ADP-bound state has been experimentally determined; however, molecular dynamics (MD) simulations and Förster Resonance Energy Transfer (FRET) analyses suggest that ATP and

**Figure 1.** Structure of SecA-SecYEG complex and HDX-MS workflow. (a) Structure and sub-domains of SecA-SecYEG (based in 3DIN as in *Zimmer et al., 2008*, modelled with a pre-protein; *Corey et al., 2016a*). The pre-protein, ATP and signal sequence are highlighted in grey, orange and yellow, respectively. (b) Top view of the complex highlighting the nucleotide-binding-domains NBD1 and NBD2 as well as PPXD (Pre-Protein cross-linking Domain) and HWD (Helical Wing Domain) (c) Overview of the HDX-MS process. The sample is prepared in detergent micelles (DDM) and after addition of nucleotides is incubated in a deuterated solvent (1). Following deuteration of the mixture at different time-points, the HDX reaction is quenched and the protein is digested with pepsin (2). Peptides are separated by liquid chromatography and subsequently identified by mass spectrometry (MS). The mass uptake of the protein in different conditions (e.g. AMPPNP vs ADP) is then compared (3). Peptides with significant difference in deuterium uptake are mapped onto a Woods plot (4). The length of lines represent the peptide sizes. Blue and red regions indicate significant protection and deprotection, respectively. Insignificant differences, calculated by a 99% confidence interval, are shown in grey.
DOI: https://doi.org/10.7554/eLife.47402.002

ADP respectively favour open and closed forms of the SecY channel (*Fessl et al., 2018*; *Allen et al., 2016*). This behaviour was incorporated into a 'Brownian ratchet' model to describe the mechanism of ATP driven protein translocation, whereby ATP binding and hydrolysis act to bias pre-protein diffusion in an outward direction. This directionality is further augmented by other factors, including the promotion of pre-protein folding on the outside, but not the inside (*Corey et al., 2019*), and by coupling transport to the trans-membrane proton-motive-force (PMF) (*Brundage et al., 1990*).

One criticism of the Brownian ratchet model is the effect of non-hydrolysable ATP analogues: while they cause the channel through SecY to open-up (*Allen et al., 2016*), they have also been shown to prevent backsliding of trapped translocation intermediates (*Bauer et al., 2014*; *Erlandson et al., 2008*). If the pre-protein channel opens in response to ATP, one might reasonably

expect backsliding to occur more readily – not less. To explain this apparent contradiction, one could invoke a second clamp that tightens when ATP is bound and prevents backwards diffusion. Indeed, SecA is known to form a clamp for the translocating pre-protein (*Zimmer et al., 2008*), so it could plausibly fulfil this role. Thus far, however, experimental evidence for this idea is lacking. Furthermore, as the SecA clamp can be fixed in a closed state without preventing transport (*Gold et al., 2013*), the conformational changes involved are likely to be subtle, and not necessarily captured with techniques such as FRET.

Clearly, the key to understanding the mechanism that prevents unfavourable back-sliding and refinement of the Brownian ratchet model is to learn more about the dynamic action of SecA-SecYEG at different stages of the ATP hydrolytic cycle. In this study, we exploit recent technical advances in HDX-MS to probe the structural differences between the ATP and ADP associated states that underpin function.

HDX-MS has emerged as a non-invasive and highly sensitive method for interrogating the conformational dynamics of membrane proteins and their complexes (*Adhikary et al., 2017*; *Eisinger et al., 2017*; *Martens et al., 2018*; *Reading et al., 2017*). HDX-MS measures the exchange of backbone amide hydrogen to deuterium at peptide-level resolution (*Englander and Kallenbach, 1983*; *Konermann et al., 2011*; *Engen, 2009*). The rate of HDX depends on the solvent accessibility, protein flexibility, and hydrogen bonding networks (*Mistarz et al., 2016*). Thus, HDX-MS can monitor the conformational changes of protein complexes by comparing HDX rates between distinct protein states (e.g.. protein *versus* protein with ligand). Consequently, comparative HDX-MS has emerged as an attractive tool to define conformational changes to complement other, high-resolution techniques. For example, the analysis of the structural consequences of protein-ligand/drug interactions, nucleotide dependent protein dynamics; ATP/ GTP *versus* ADP/ GDP (*Murcia Rios et al., 2018*; *Narang et al., 2018*); including those associated with membrane proteins and complexes (*Adhikary et al., 2017*; *Martens et al., 2018*; *Reading et al., 2017*). Here, we use HDX-MS to systematically interrogate nucleotide-induced conformational changes in the multi-component SecA-SecYEG complex.

## Results

To investigate the conformational dynamics of SecA, we prepared samples of the enzyme either alone or saturated with an excess of purified SecYEG. We used HDX-MS to study the conformational dynamics of the complex. In each case, the experiments were run in the presence of ADP or a non-hydrolysable analogue of ATP (AMPPNP), and only the data for the saturated component (SecA) was used. AMPPNP was used to study the ATP-bound state because ATP is rapidly hydrolysed to ADP in the active complex: all mentions hereafter of ATP pertinent to the HDX-MS experiments refer to AMPPNP.

Optimised procedures for HDX-MS were carried out as described previously (*Corey et al., 2019*; *Martens et al., 2018*) (*Figure 1c*). The peptide coverage of the whole complex was high (ranging from 82% to 95% for the respective subunits; *Figure 2—figure supplement 1*), enabling a thorough analysis of SecA and SecYEG. The comparison of the AMPPNP *versus* ADP datasets produced a differential HDX-MS fingerprint (ΔHDX) that allowed direct comparisons between the results obtained for distinct states (AMPPNP - ADP; *Figure 1c*). To interpret the data, we utilised the Woods plot, which displays the length and the beginning and end residues of each analysed peptide against their respective ΔHDX. Woods plots were prepared using our in-house software (Deuteros; *Lau et al., 2019*).

We began by investigating the influence of nucleotides on the conformational dynamics of SecA. ΔHDX experiments were performed allowing us to track structural changes between AMPPNP and ADP bound states (SecA$^{AMPPNP}$ – SecA$^{ADP}$). Surprisingly, there was very little effect on the conformational dynamics of the isolated SecA (*Figure 2a,b* and *Figure 2—figure supplement 2a*). The greatest difference in deuterium uptake between the two states was observed at and around the ATP-binding sites (*Figure 2a,b*). This highlights the fact that SecA is only activated when it is associated with SecYEG: studying SecA by itself is therefore unlikely to yield information pertinent to transport.

To further explore the mechanism of protein transport we carried out equivalent comparative HDX-MS experiments in the presence of an excess of SecYEG to saturate SecA (SecA$^{AMPPNP}$SecYEG - SecA$^{ADP}$SecYEG) (*Figure 2a,c* and *Figure 2—figure supplement 2b*). The results are consistent

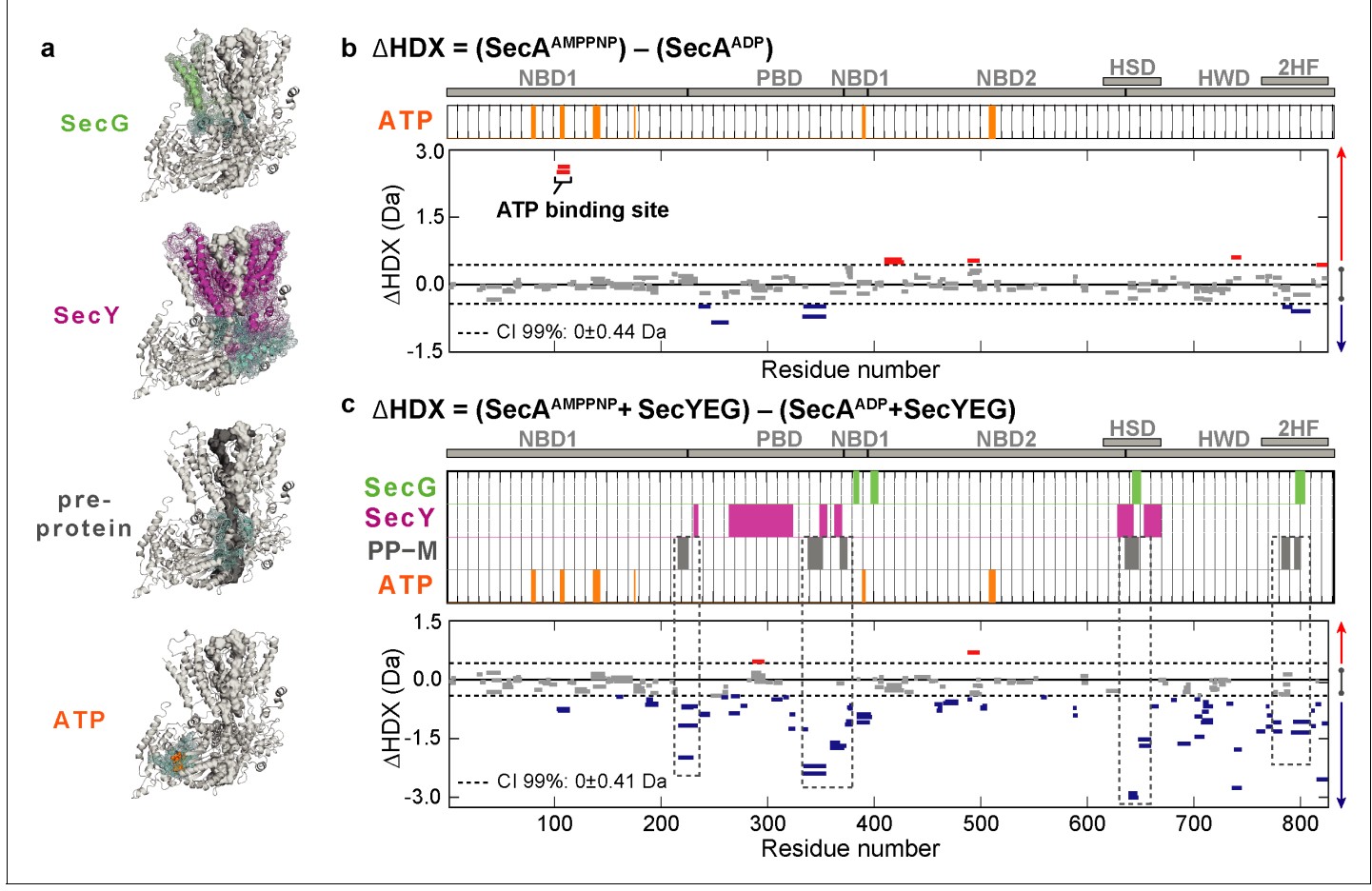

**Figure 2.** Impact of nucleotides on the conformational dynamics of SecA. (a) Structures of the complex highlighting the contact sites (cyan) in SecA with SecG (green), SecY (pink), mature regions of the pre-protein (PP-M; grey) and ATP (orange). Significant sum differences in relative deuterium uptake (ΔHDX = AMPPNP-ADP) of (b) SecA without SecYEG, and (c) of SecA in the presence of a molar excess of SecYEG after 30 min of deuteration. Highlighted regions represent contacts with SecG, SecY, pre-protein (mature domain, PP-M) and ATP; coloured according to (a). Dashed boxes in (c) highlight regions in contact with the pre-protein mature domain (PP-M). Detailed information of HDX-MS data is provided in *Supplementary files 1a and 1b*.

DOI: https://doi.org/10.7554/eLife.47402.003

The following figure supplements are available for figure 2:

**Figure supplement 1.** Linear protein coverage maps for SecA, SecY, SecE and SecG.

DOI: https://doi.org/10.7554/eLife.47402.004

**Figure supplement 2.** Woods plots illustrating the impact of nucleotides on the conformational dynamics of SecA.

DOI: https://doi.org/10.7554/eLife.47402.005

**Figure supplement 3.** Impact of SecYEG binding on SecA.

DOI: https://doi.org/10.7554/eLife.47402.006

with SecYEG having a profound effect on SecA: SecYEG induces a large ATP-dependent stabilisation across almost the entirety of SecA (*Figure 2c*; negative ΔHDX, blue bars). We also examined the conformational effects of SecYEG binding to SecA (*Figure 2—figure supplement 3*). Interestingly, the results suggested small differences as compared to the differences observed between ATP- and ADP-bound states of SecA whilst bound to SecYEG (*Figure 2c* and *Figure 2—figure supplement 3*). This implies that the presence of nucleotides conformationally activate SecA.

For mechanistic interpretation, the data were analysed in the context of a model of the SecA-SecYEG bound to a short pre-protein (*Corey et al., 2016a*)(*Figure 2a*). Most striking was the correspondence of the stabilised regions with sites in SecA that directly contact the mature regions (PP-M) of the translocating pre-protein (*Figure 2a,c*; dashed boxes –PP-M, grey bars). Such stabilisation

in SecA suggests an ATP-driven conformational closure around the mature segment of the pre-protein. Conversely, of course, the highlighted regions identified in SecA would be destabilised in the ADP-bound state, following ATP hydrolysis.

To explore the nucleotide-dependent conformational dynamics of the complex, we compared the behaviour of SecA to the associated channel complex SecYEG. Previously, we conducted ΔHDX analysis of SecA-SecYEG, as above, but instead saturated SecYEG with SecA (*Corey et al., 2019*). In that instance, we were only interested in selecting peptides lining the SecY protein-channel. Here, by taking into account global ΔHDX (SecA$^{AMPPNP}$SecYEG - SecA$^{ADP}$SecYEG), we highlight the long-range nucleotide dependent impact of SecA on the dynamics of SecYEG (30 min deuterium exchange; not previously shown; *Figure 3*; *Corey et al., 2019*).

Interestingly, we found significant ATP-dependent destabilisation of SecY (*Figure 3a,b*, and *Figure 3—figure supplement 1*). Closer inspection of the identified peptides reveals that two prominently destabilised regions in the SecY protein would be in direct contact with the signal sequence (PP-S) and mature regions (PP-M) of the pre-protein during protein translocation (*Figure 3b*; dashed boxes –PP-M grey, and PP-S yellow bars). These regions include the plug, which in the absence of the pre-protein maintains the closed state of the channel (*Van den Berg et al., 2004*), and the highly conserved trans-membrane helix 7. Multiple regions within the SecE and SecG subunits were also significantly destabilised in the presence of AMPPNP (*Figure 3a,c–d* and *Figure 3—figure supplement 1*). Again, conversely those identified regions would be stabilised after hydrolysis to ADP. These observations are consistent with an ATP-driven opening of the SecY channel, and closure after hydrolysis (*Fessl et al., 2018*; *Allen et al., 2016*).

The ΔHDX analysis describes contrasting behaviour between the protein channel and cytoplasmic motor components of the bacterial translocon, during the ATP binding and hydrolysis cycle. Notably, there is an ATP-dependent closure around the translocating polypeptide by SecA and opening of the channel through SecYEG (*Figure 4a*); reversed in the ADP bound state after ATP hydrolysis (*Figure 4b*). To further understand this effect with respect to local structural rearrangements in SecA, we re-analysed previously-run all-atom molecular dynamics (MD) simulations of the SecA-SecYE complex engaged with a stretch of pre-protein (*Corey et al., 2019*). The size of two distinct pores in the SecA pre-protein channel through which the pre-protein passes were quantified (*Figure 4c,d*; see also Materials and methods for details); note that both pores contain residues shown by HDX-MS to be protected in the ATP-bound state (*Figure 2*, *Figure 4c* and Materials and methods).

From the simulation data, it is clear that both SecA pores, particularly pore one at the cytoplasmic entrance of SecA, are more constricted in the ATP-bound state (*Figure 4c,d*). This is consistent with the HDX data and suggests that ATP-binding is causing SecA to close around the engaged pre-protein.

## Discussion

Overall, our results reveal profound nucleotide-dependent differences in the structure of the active Sec complex (SecA-SecYEG). These differences can broadly be divided into two groups: those affecting the membrane channel (SecY, SecE and SecG), which become destabilised in the presence of ATP, and those affecting SecA, which become stabilised. A more detailed analysis reveals that the most dramatic changes affect regions that would directly contact a translocating pre-protein. These results are consistent with the previously proposed Brownian ratchet mechanism of pre-protein transport (*Allen et al., 2016*). The cyclical ATP/ADP-dependent conformational dynamics highlighted in SecA and SecYEG are presumably integral components of the underling mechanism of this ratcheting machine, for instance, the opening (ATP) and closure (ADP) of the channel through SecY (*Corey et al., 2019*). Secondly, the conferral of ATP-dependent asymmetry that prevents pre-protein folding at the cytosolic side of the membrane, while allowing it at the exterior (*Corey et al., 2019*) is likely to also be a result of these changes. Finally, the observation that ATP causes SecA to constrict around the pre-protein could potentially explain why pre-proteins do not backslide in the presence of a non-hydrolysable ATP analogue (*Bauer et al., 2014*; *Erlandson et al., 2008*); and thereby promote forward transport.

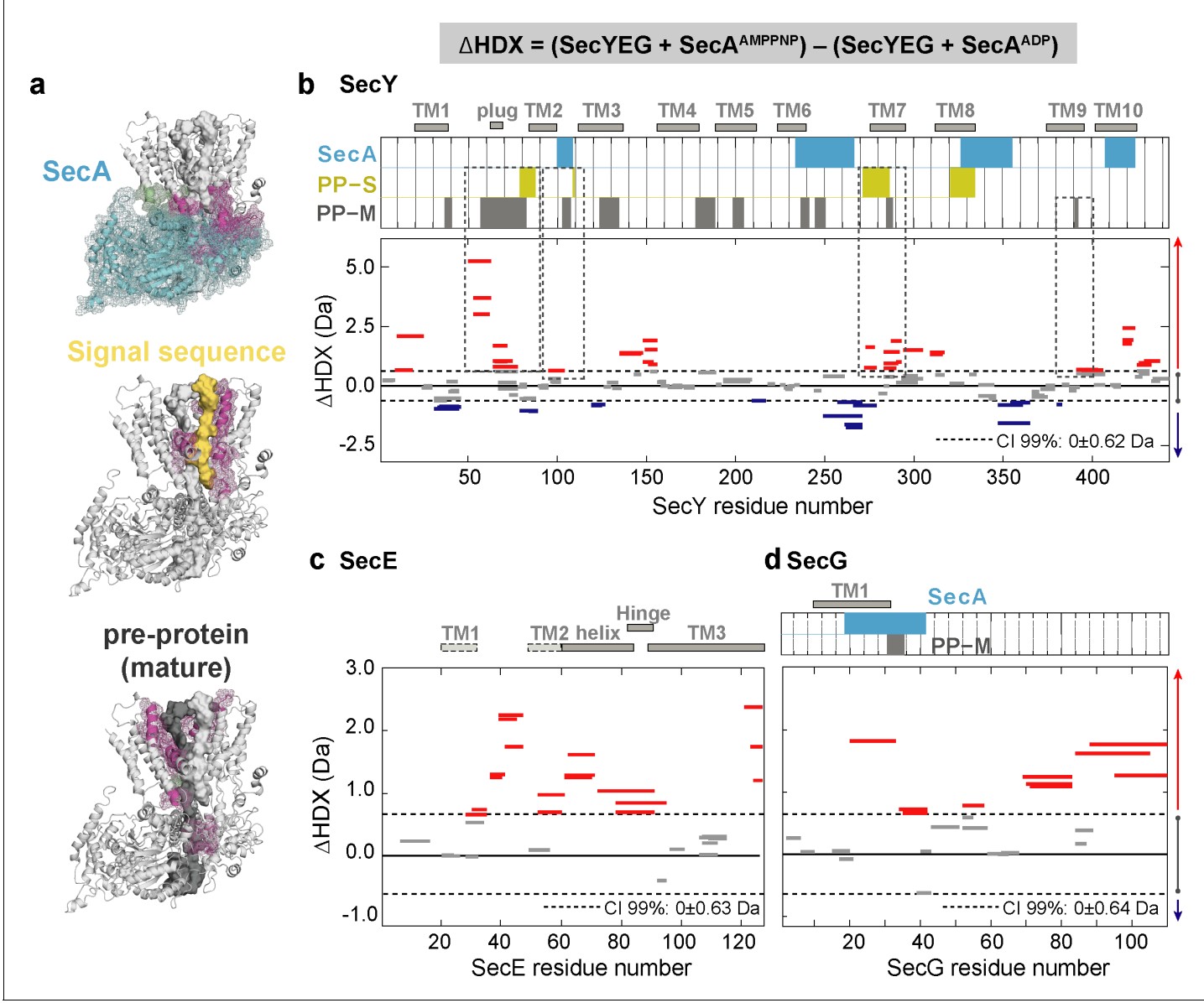

**Figure 3.** Impact of nucleotides on the conformational dynamics of SecY, SecE and SecG. (a) Structures of the SecA-SecYEG complex highlighting the contact sites of SecY (pink) and SecG (green) with SecA (cyan), signal sequence (PP-S; yellow) and pre-protein (PP-M; grey). (b-d) Significant sum differences in relative deuterium uptake (ΔHDX = AMPPNP-ADP) of (b) SecY, (c) SecE and (d) SecG in the presence of excess SecA at 30 min of deuteration. Regions interacting with SecY (b) and SecG (d) are highlighted: SecA, pre-protein mature domain (PP-M) and pre-protein signal sequence (PP-S); colour coding as in (a). Confidence intervals (CI; 99%) are shown as grey dotted lines. Red and blue bars indicate structural stabilisation (positive ΔHDX) and destabilisation (negative ΔHDX) of peptides, respectively. Grey bars indicate peptides with insignificant ΔHDX. Dashed boxes in (b) highlight regions interacting with the pre-protein signal sequence (PP-S) and the mature pre-protein (PP-M). Detailed information of HDX-MS data is provided in *Supplementary file 2*.

DOI: https://doi.org/10.7554/eLife.47402.007

The following figure supplement is available for figure 3:

**Figure supplement 1.** Woods plots illustrating the impact of nucleotides on the conformational dynamics of SecY, SecE and SecG.

DOI: https://doi.org/10.7554/eLife.47402.008

The experiments presented here demonstrate the power of combining HDX-MS with the results of biochemistry and molecular simulations to understand the conformational mechanism of multi-component membrane protein complexes, difficult to study by other approaches.

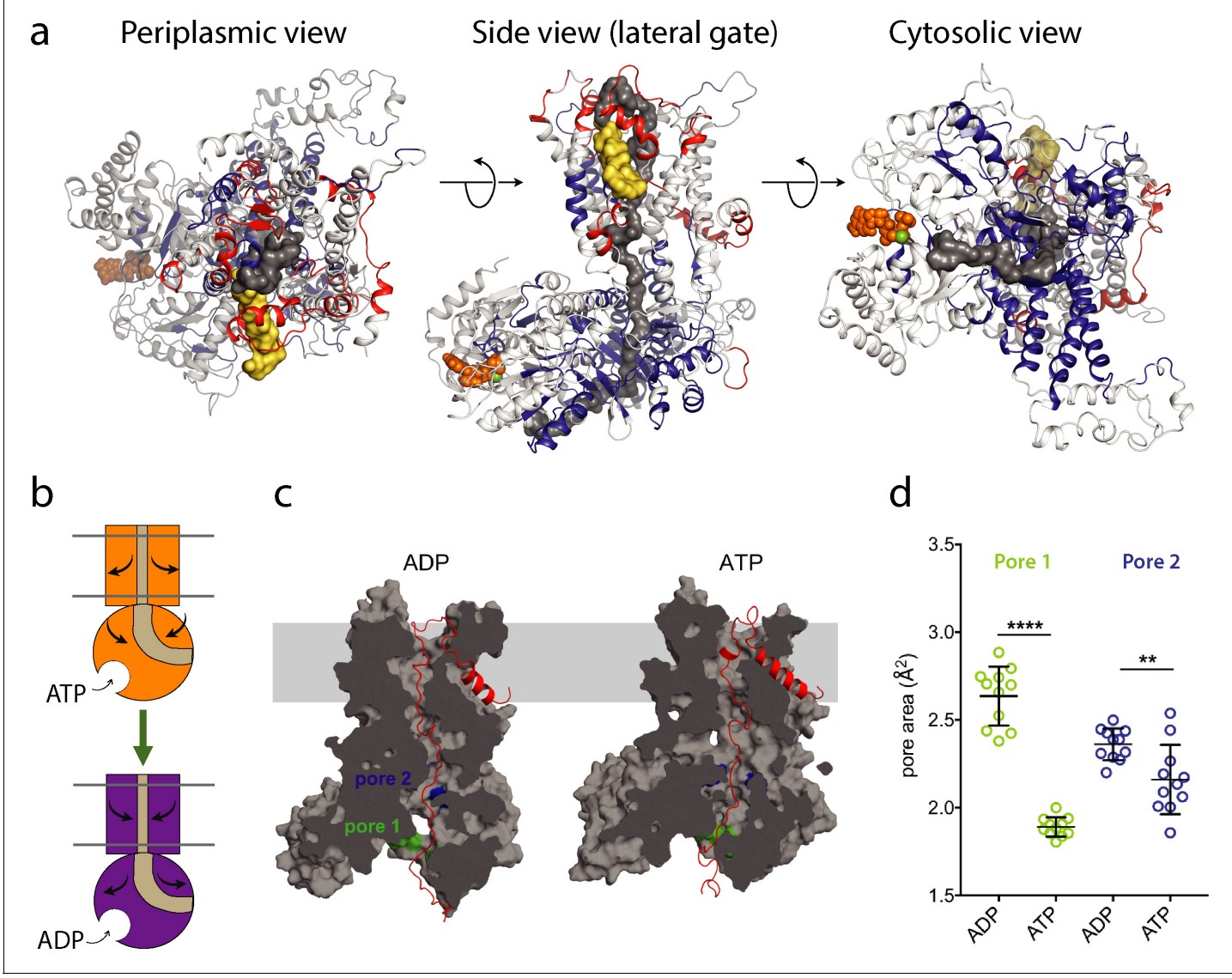

**Figure 4.** ATP-regulation of channel size. (**a**) Periplasmic, membrane perpendicular and cytosolic views of the SecA-SecYEG-preprotein complex model, with ΔHDX of the peptides mapped onto the structure. The translocon complex is coloured light grey, with H-bonding destabilised and stabilised regions in red and blue, respectively. The pre-protein is shown in yellow (signal sequence) and dark grey (mature). ATP is shown as orange spheres. (**b**) Schematic summarising the primary outcomes from the data here. (**c**) Snapshots of SecA-SecYE with an engaged pre-protein after 1 μs all-atom MD simulation (*Corey et al., 2019*). SecA-SecYE is shown as grey surface and has been slabbed to show the pre-protein channel through both SecA and SecY. The pre-protein is shown as red cartoon, and was absent for the cavity size analysis in panel (**d**). The positions of the conserved SecA pores have been highlighted; pore 1 (green) and pore 2 (blue). Visual analysis suggests that pore 1 is more constricted in the ATP state (**d**). Quantification of the pore size, using snapshots every 25 ns from 750 to 1000 ns: both pores are tighter in ATP bound state (pore 1 means are 1.9 and 2.6 Å², p<0.0001 using a 2 tailed *t*-test: pore 2 means are 2.4 and 2.1 Å², p=0.0061).

DOI: https://doi.org/10.7554/eLife.47402.009

## Materials and methods

### Protein preparation

SecYEG and SecA were overproduced and purified as described previously (*Gold et al., 2007*).

## Hydrogen deuterium exchange mass spectrometry (HDX-MS)

The HDX-MS experiments outlined here were carried out using a Synapt G2-Si HDMS coupled to an Acquity UPLC M-Class system with HDX and automation (Waters Corporation, Manchester, UK).

To form the SecA-SecYEG complex, SecYEG and SecA were mixed and incubated for 10 min on ice. For experiments investigating SecA (*Figure 2*), SecA (10 μM) was saturated by SecYEG (15 μM). For experiments investigating SecYEG (*Figure 3*), SecA (15 μM) was added in excess of SecYEG (10 μM), as described previously (*Corey et al., 2019*). 5 μl of SecA or the SecA-SecYEG complex was diluted into 95 μl of deuteration buffer (20 mM Tris pH 8, 2 mM $MgCl_2$, 50 mM KCl, and 0.02% (w/v) dodecyl-maltoside (DDM) in $D_2O$) or with equilibration buffer (20 mM Tris pH 8, 2 mM $MgCl_2$, 50 mM KCl, and 0.02% (w/v) DDM in $H_2O$). Deuteration was carried out at 25°C. For experiments analysing the Sec complex in the presence of the nucleotides, 1 mM of either the non-hydrolysable AMPPNP or ADP was added to the protein mixture and to equilibrium or deuteration buffer.

The proteins were labelled by incubation in deuteration buffer for 0.25, 1, 5, and 30 min to capture short, medium and long exchange times. Deuteration was quenched with 100 μL of quench buffer (0.7% (v/v) formic acid and 0.1% (w/v) DDM) at 1°C and a pH of 2.4. Protein digestion to peptides was performed at 25°C using an Enzymate online digestion column (Waters) in 0.1% (v/v) formic acid at a flow rate of 200 μL/min. Between injection of samples, the pepsin column was washed with cleaning solution (0.8% (v/v) formic acid, 1.5 M Gu-HCl, and 4% (v/v) MeOH) recommended by the manufacturer. To reduce peptide carry-over, a blank run was performed between sample runs.

Peptides were trapped using an Acquity BEH C18 1.7 μM VANGUARD pre-column for 3 min at a flow rate of 200 μL/min in buffer A (0.1% (v/v) formic acid at a pH of ~2.5). Peptides were eluted into an Acquity UPLC BEH C18 1.7 μM 1.0 × 100 mm analytical column with a linear gradient of 8–40% (v/v) gradient of acetonitrile with 0.1% (v/v) formic acid at a flow rate of 40 μL/min. Peptides were then ionised by positive electrospray into a Synapt G2-Si mass spectrometer (Waters). A 20–30 V trap collision energy ramp was utilised to capture the $MS^E$ data. The electrospray ionisation source was operated in a positive ion mode and ion mobility was enabled for all experiments. Leucine Enkephalin was used as a lock mass for mass accuracy correction and iodide was used for mass spectrometry calibration. All deuterium time points were performed in triplicate.

## HDX data evaluation and statistical analysis

All experiments, including deuterated time points and reference sample controls, were repeated in triplicate. $MS^E$ data from reference sample controls of the complexes were used by the Waters ProteinLynx Global Server 2.5.1 (PLGS) and filtered using DynamX (v. 3.0) to provide sequence identification. The following parameters were used to filter the quality of the peptides: minimum and maximum peptide sequence length of 4 and 25, respectively, minimum intensity of 1000, minimum MS/MS products of 2, minimum products per amino acid of 0.2, and a maximum MH + error threshold of 5 ppm. All spectra generated from the peptides were examined and only peptides with high-quality spectra and a high signal to noise ratio were used for data analysis. Wood's plots and confidence intervals were generated using the in-house Deuteros software (*Lau et al., 2019*).

## *In silico* analyses of SecA pore dynamics

Analyses were based on all-atom molecular dynamics simulations of a complex comprising *Bacillus subtilis* SecA, *Geobacillus thermodenitrificans* SecYE and a 76-stretch of pre-protein, built from PDB 5EUL (*Li et al., 2016*). The simulations were run in an ATP- or ADP-bound state, with full details of their set up described previously (*Corey et al., 2019*). For the analyses here, only the atoms corresponding to SecA and SecYE were kept (i.e. the solvent, membrane and pre-protein were removed). Snapshots were taken every 25 ns over a range of 750 to 1000 ns.

Two highly conserved pre-protein pores in SecA were identified: pore 1 on the cytoplasmic surface of SecA (*Figure 4c*: green) and pore 2 close to the SecY binding interface (*Figure 4c*: blue). Pore 1 consists of residues Ile 222, Ser 224, Gly 326, Arg 327, Arg 328 and Ser 340 (in *B. subtilis* numbering; equivalent to *E. coli* residues Ile 224, Ser 226, Gly 346, Arg 347, Arg 348 and Ser 350). Pore 2 consists of Gln 595, Tyr 599 and Gln 736 on SecA (in *B. subtilis* numbering, *E. coli* equivalent are Gln 644, Tyr 648 and Gln 787), and Ile 243, Tyr 245 and Ala 246 on the functionally important (*Corey et al., 2016b*) C4 loop of SecY (*G. thermodentrifinicans* numbering, approximately equivalent to Val 246, Tyr 248 and Ala 249 on *E. coli* SecY). Note that residues Gly 346, Arg 347, Arg 348

and Ser 350 in pore one and Gln 787 and Ala 249 of pore 2 (all in *E. coli* numbering) are shown to be deprotected in the HDX data (see Data Availability).

To quantify the pore size in the different snapshots, cavity cross-sectional area analyses were run using HOLE (*Smart et al., 1996*). The algorithm was set to start from a position in either pore 1 (between the α-carbons of residues Ile 222 and Ser 340) or pore 2 (between the α-carbons of residues Tyr 599 and Ile 243). To account for flexibility in the pore position, a region of ~0.9 nm on either side of the starting point was selected for analysis, and the pore defined as the narrowest point in this region. The pore size was quantified based on the average over a 0.35 nm window, to reduce artefacts arising from thermal fluctuations.

Images were made using the PyMOL Molecular Graphics System version 2.1.1, Schrödinger, LLC, and data were plotted and analysed in Prism version 7, GraphPad Software.

## Data availability

The mass spectrometry proteomics data have been deposited to the ProteomeXchange Consortium via the PRIDE (*Vizcaíno et al., 2016*) partner repository with the dataset identifier: PXD013594.

SecYEG-SecA (saturated SecYEG from Corey et al., eLife 2019;8:e41803 DOI: 10.7554/eLife.41803) data can be accessed using the with the dataset identifier: PXD013594.

## Acknowledgements

This work was supported by the Wellcome Trust (109854/Z/15/Z) and a King's Health Partners R and D Challenge Fund through the MRC, Confidence in Concept (MC_PC_15031) to AP. This work was funded by the BBSRC (BB/N015126/1 to IC; BB/M003604/1 to IC and RAC; BB/I008675/1 to IC and WJA). Extended simulations were run on the ARCHER UK National Supercomputing Service (http://www.archer.ac.uk), provided by HECBioSim, the UK High End Computing Consortium for Biomolecular Simulation (hecbiosim.ac.uk), supported by the EPSRC. EP is the recipient of an Imperial College London, Institute of Chemical Biology EPSRC CDT studentship.

## Additional information

### Funding

| Funder | Grant reference number | Author |
| --- | --- | --- |
| Wellcome | 109854/Z/15/Z | Argyris Politis |
| Medical Research Council | MC_PC_15031 | Argyris Politis |
| Biotechnology and Biological Sciences Research Council | BB/N015126/1 | Ian Collinson |
| Biotechnology and Biological Sciences Research Council | BB/M003604/1 | Ian Collinson |
| Biotechnology and Biological Sciences Research Council | BB/I008675/1 | Ian Collinson |

The funders had no role in study design, data collection and interpretation, or the decision to submit the work for publication.

### Author contributions

Zainab Ahdash, Euan Pyle, Data curation, Formal analysis, Validation, Investigation, Visualization, Methodology, Writing—review and editing; William John Allen, Data curation, Formal analysis, Investigation, Visualization, Writing—review and editing; Robin A Corey, Data curation, Investigation, Visualization, Writing—review and editing; Ian Collinson, Argyris Politis, Conceptualization, Supervision, Funding acquisition, Writing—original draft, Project administration, Writing—review and editing

## Author ORCIDs

Zainab Ahdash (iD) https://orcid.org/0000-0002-4495-8689
Euan Pyle (iD) https://orcid.org/0000-0002-4633-4917
William John Allen (iD) http://orcid.org/0000-0002-9513-4786
Robin A Corey (iD) https://orcid.org/0000-0003-1820-7993
Ian Collinson (iD) https://orcid.org/0000-0002-3931-0503
Argyris Politis (iD) https://orcid.org/0000-0002-6658-3224

## Decision letter and Author response

Decision letter https://doi.org/10.7554/eLife.47402.016
Author response https://doi.org/10.7554/eLife.47402.017

# Additional files

## Supplementary files

• Supplementary file 1. HDX-MS data summary table for $\Delta HDX$ = (SecA$^{AMPPNP}$) – (SecA$^{ADP}$). (**A**) Data is presented in *Figure 2b*. (**B**) HDX-MS data summary table for $\Delta HDX$ = (SecA$^{AMPPNP}$ +SecYEG) – (SecA$^{ADP}$ +SecYEG). Data is presented in *Figure 2c*. Mass spectrometry raw data are deposited in the ProteomeXchange Consortium via the PRIDE database.
DOI: https://doi.org/10.7554/eLife.47402.010

• Supplementary file 2. HDX-MS data summary table for $\Delta HDX$ = (SecYEG +SecA$^{AMPPNP}$) – (SecYEG +SecA$^{ADP}$). Data is presented in *Figure 3*. Mass spectrometry raw data are deposited in the ProteomeXchange Consortium via the PRIDE database.
DOI: https://doi.org/10.7554/eLife.47402.011

• Transparent reporting form
DOI: https://doi.org/10.7554/eLife.47402.012

## Data availability

All data generated or analysed during this study are included in the manuscript and as supplementary figures and tables. Raw HDX-MS proteomics data have been deposited to the ProteomeXchange Consortium via the PRIDE31 partner repository with the dataset identifier: PXD013594.

The following dataset was generated:

| Author(s) | Year | Dataset title | Dataset URL | Database and Identifier |
|---|---|---|---|---|
| Ahdash Z, Pyle E, Allen WJ, Corey RA, Collinson I, Politis A | 2019 | HDX-MS data from: HDX-MS reveals nucleotide-dependent, anti-correlated opening and closure of SecA and SecY channels of the bacterial translocon | http://proteomecentral. proteomexchange.org/ cgi/GetDataset?ID= PXD013594 | ProteomeXchange Consortium, PXD0 13594 |

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
