## [Decision Letter]

Thank you for submitting your article "HDX-MS reveals nucleotide-based, anti-correlated opening and closure of SecA/ SecY channels of the bacterial translocon" for consideration by *eLife*. Your article has been reviewed by three peer reviewers, one of whom is a member of our Board of Reviewing Editors, and the evaluation has been overseen by Richard Aldrich as the Senior Editor. The reviewers have opted to remain anonymous.

The reviewers have discussed the reviews with one another and the Reviewing Editor has drafted this decision to help you prepare a revised submission.

Summary:

Using hydrogen-deuterium exchange experiment and molecular dynamics simulations, this work investigated and provided important insight into the structural-mechanism of protein translocation by SecA. The finding is consistent with the previously proposed Brownian ratchet mechanism in showing channel-opening of SecY and simultaneous channel-tightening of SecA upon ATP binding of the latter.

Essential revisions:

While the reviewers were satisfied with the quality of the data and found the results of this work potentially interesting and significant, they believe that the conclusion can be further strengthened by showing that the altered pattern of hydrogen-deuterium exchange upon ATP binding is indeed a result of SecA tightening and SecY opening upon ATP binding. As is, this is not necessarily the case because other changes in conformational dynamics such as disruption of protein secondary structure could also alter the HD exchange pattern without SecA tightening and SecY opening. The reviewers suggest that further analysis on the MD data is required to show that from the MD data on can qualitatively predict the HDX patterns. This could be done in principle using previously reported methods such as that by Park et al. (J. Chem. Inf. Model. 2015, 55, 1914−1925).

The authors should expand the Discussion somewhat to contextualize their results with respect to the current understanding of SecY translocation mechanisms.

---

## [Author Response]

Essential revisions:While the reviewers were satisfied with the quality of the data and found the results of this work potentially interesting and significant, they believe that the conclusion can be further strengthened by showing that the altered pattern of hydrogen-deuterium exchange upon ATP binding is indeed a result of SecA tightening and SecY opening upon ATP binding. As is, this is not necessarily the case because other changes in conformational dynamics such as disruption of protein secondary structure could also alter the HD exchange pattern without SecA tightening and SecY opening. The reviewers suggest that further analysis on the MD data is required to show that from the MD data on can qualitatively predict the HDX patterns. This could be done in principle using previously reported methods such as that by Park et al. (J. Chem. Inf. Model. 2015, 55, 1914−1925.).

We carried out the suggested additional analysis showing that the MD data can indeed qualitatively predict the HDX patterns (Author response image 1). In particular, we have constructed a scoring function for the MD data, based partly on the Park et al. paper mentioned above, but also on Markwick et al. [Biophys J. 2019 Jan 8; 116(1): 49–56]. In the latter, it has been shown that the theoretical predictions best matched experiments when both H-bonding and solvent accessibility statistics were considered. Our theoretical prediction considered both such statistics as follows.

In short, each backbone amine was given an approximate HDX score based on the MD data – if hydrogen bonded to another protein atom it receives a score of 0. If not, it receives a score of 0 or 1 based on non-zero exposure to solvent, as determined using solvent accessible surface area analysis. We computed this value for each backbone amine in SecA in the ATP-bound state at 6 different time points (0.95, 0.96, 0.97, 0,98, 0,99 and 1 µs). We summed the average values for the residues in each equivalent peptide from the HDX analysis (with the first residue removed, as described in Park et al.).

We plot the experimental HDX uptake vs. the simulated uptake in Author response image 1. There is a good correlation between the data (P < 0.0001 on F-test of slope being different from zero, R^2^ = 63%). We have not extended the analyses to SecYEG, as the difference in membrane environment and extra SecE TMs in *E. coli* will likely make meaningful analysis difficult.

It is worth noting that for the present study, we feel that a more detailed, quantitative comparison of the HDX and MD is made difficult by the differences in sequence (*Thermotoga* vs. *E. coli*) – in particular, there are a considerable number of extra regions on the surface of each system that will undoubtedly affect exchange. Moreover, there is considerable mismatch between the timescales of MD (µs) and HDX-MS experiments (s to min). Nevertheless, our analysis provides a qualitatively correlation between the MD simulations and HDX-MS experiments.

**Author response image 1. respfig1:** Experimental HDX uptake vs. theoretically predicted uptake; the theoretical uptake was constructed by taking into account both H-bonding and solvent accessibility statistics.

The authors should expand the Discussion somewhat to contextualize their results with respect to the current understanding of SecY translocation mechanisms.

The Discussion includes a short statement, now extended, to this effect. It is succinct and to the point, appropriate for a short report and the allowable length.